# Plant Growth Regulators Improve Grain Production and Water Use Efficiency of *Foeniculum vulgare* Mill. under Water Stress

**DOI:** 10.3390/plants11131718

**Published:** 2022-06-28

**Authors:** Ghasem Parmoon, Ali Ebadi, Masoud Hashemi, Barbara Hawrylak-Nowak, Carol Baskin, Soodabe Jahanbakhsh

**Affiliations:** 1Department of Agronomy, Faculty of Agriculture, University of Mohaghegh Ardabili, Ardabil P.O. Box 179, Iran; ebadi@uma.ac.ir (A.E.); jahanbakhsh@uma.ac.ir (S.J.); 2Stockbridge School of Agriculture, University of Massachusetts, 207 Bowditch Hall, 201 Natural Resources Road, Amherst, MA 01003, USA; masoud@umass.edu; 3Department of Botany and Plant Physiology, Faculty of Environmental Biology, University of Life Sciences in Lublin, Akademicka 15, 20-950 Lublin, Poland; barbara.nowak@up.lublin.pl; 4Department of Biology, Plant and Soil Sciences, University of Kentucky, Lexington, KY 40546, USA; carol.baskin@uky.edu

**Keywords:** epibrassinolide, fennel, *Foeniculum vulgare* Mill., grain yield, methyl jasmonate, putrescine, water use efficiency, water stress

## Abstract

The development of methods increasing plant water use efficiency (WUE) would enhance the ability to grow wild aromatic and medicinally important species. The aim of this research was to determine the effect of plant growth regulators (PGRs) applied by spraying on stress resistance and WUE of fennel subjected to water stress. Plants in the generative stage were more drought tolerant than those in the vegetative stage. Water stress at vegetative stage decreased plant biomass and grain yield by 60% and 61%, respectively. Severe water stress in vegetative stage reduced grain production by 56%, and grains had 43% lower mass than those from non-stressed plants. Application of PGRs at both stages of growth increased grain yield and biomass, but the magnitude of increase depended on the type and application time of PGRs. Plants grown in well-watered conditions and sprayed with methyl jasmonate during the vegetative stage had the highest grain production (2.7 g plant^−1^), whereas under moderate water stress, plants yielded the best (2.1 g plant^−1^) when sprayed with epibrassinolide. The maximum WUE for grain (0.91 g L^−1^) and essential oil production (20 mg L^−1^) was noted in plants exposed to moderate stress and treated with methyl jasmonate during the vegetative stage.

## 1. Introduction

Water stress can cause serious alterations in many physiological processes, which negatively affects the growth and development of agricultural crops [1]. Plants have various defense mechanisms that alleviate the negative impact of drought conditions, including increased net photosynthesis, carbohydrate transport [2], photosynthetic enzyme gene expression [3], antioxidant enzyme activity [4], stomatal conductance, and CO_2_ assimilation rate [5,6]. The ability of plants to tolerate water stress varies with the species, stage of development, and duration and intensity of drought [7]. The development of methods to increase water use efficiency (WUE) and thus plant resistance to drought would enhance our ability to cultivate some edible and medicinally important species, such as fennel, as a crop in arid and semi-arid areas. Therefore, more information is needed on plant responses to (1) water stress in different development stages and (2) plant growth regulators (PGRs) applied at different levels of water stress and during different growth stages.

Most activities of plant cells are regulated by phytohormones and growth regulators, and these compounds can increase plant tolerance to biotic and abiotic stresses [8]. For example, brassinosteroids, a class of phytohormones with a poly-hydroxyl lactone steroidal structure, regulate different biological activities such as seed germination; promote root growth, flowering, defoliation, maturation, and cell division; and can delay aging [9,10]. In addition, these phytohormones enhance drought resistance, facilitating the distribution of assimilates through the plant and thereby increasing productivity potential [11]. Treatment of some plants such as rice (*Oryza sativa* L.) and wheat (*Triticum aestivum* L.) with brassinosteroids has increased plant height and number and mass of grains [12,13]. In turn, polyamines (e.g., putrescine) are involved in the regulation of development and plant growth [14] and DNA synthesis [15]. They can also positively affect abiotic stress tolerance, although little is known about the regulation of the metabolic pathways at the transcriptional, translational, and post-transcriptional levels by polyamines [16]. Methyl jasmonate is another phytohormone that can promote plant growth, e.g., formation of storage organs such as potato (*Solanum tuberosum* L.) tubers, induce anthocyanin accumulation, and enhance resistance to abiotic and/or biotic stresses [17]. We chose the three PGRs (epibrassinolide, putrescine, methyl jasmonate) for our research as representatives of three classes of phytohormones, due to their proven effectiveness in mitigating the effects of abiotic stresses, including water deficit, in crops.

Some reports indicate that the application of PGRs during the vegetative stage in soybean (*Glycine max* L.) improved yield of stover (matured dried stalks). Additionally, the application of PGRs at the flower initiation stage of soybean resulted in larger grain size, more pods initiated, and increased pod length [18]. Rice and corn (*Zea mays* L.) have also shown positive responses to treatment with PGRs [19,20]. Although PGRs can contribute to drought tolerance of crops, information about their effect on the growth, osmotic adjustment, WUE value as well as essential oil level in medicinal plants is still limited.

In general, the majority of medicinal and aromatic plants are known to be tolerant of mild or even severe water stress [21,22]. Although biomass and flower yield of pennyroyal (*Mentha pulegium* L.) is potentially reduced by water stress [23], exposure of medicinal plants to mild water stress generally promotes accumulation of some metabolites that can increase WUE [24]. Fennel (*Foeniculum vulgare* Mill.) (Apiaceae), one of the oldest herbs, is an important edible, medicinal, and cosmetic plant mostly grown in arid and semi-arid regions, where the water availability is the critical factor determining plant growth and yield [25]. Fennel accumulates many valuable phytochemicals in the fruits, which are a source of essential oils containing, e.g., α-phellandrene, estragole, fenchone, and trans-anethole [26]. This oil has antispasmodic impact and hepatoprotective activity [27]. Although fennel is species native to the Mediterranean region, it is now cultivated in different parts of the world [28,29]. Iran is one of the main producers and exporters of this species, which is commercially cultivated mainly for culinary and pharmaceutical uses and occupy very different habitats. When cultivated in the central or western parts of Iran, not only is there the production of the highest yield of seeds but also the highest accumulation of essential oil and anethol [30].

We tested the hypothesis that fennel plants have high WUE, and the negative effects of water stress imposed during the vegetative or generative stages will be alleviated by using PGRs, resulting in higher grain and essential oil production that in PGR-untreated plants. Since the availability of irrigation water is the major constraint to crop production in arid climates, the detailed purpose of this study was to determine (1) the response of fennel to moderate and severe water stress and (2) the effect of application of selected PGRs on fennel water deficit tolerance (expressed by plant yielding), WUE, and accumulation of essential oil. The results of this study will be useful for fennel researchers with application potential to improve the quality and quantity of its yield under water deficit.

## 2. Materials and Methods

### 2.1. Experimental Site Characteristics

Experiments were conducted in an air-conditioned greenhouse located on the campus of the University of Mohaghegh Ardabili, Ardabil, Iran. Day and night temperatures in the greenhouse were 24 ± 2 °C and 16 ± 2 °C, respectively. Relative humidity was 50 ± 5%, and maximal photosynthetic photon flux density (PPFD) from sunlight in summer season was about 700 μmol m^−2^ s^−1^. Plants were grown individually in pots (diameter = 20 cm and height= 50 cm) that contained 20 kg of soil (loam type) collected from the field and sieved to remove stones. At the time of planting, the pH of soil (1:1, soil:H_2_O) was 7.9, cation exchange capacity (CEC) was 0.625 ds m^−2^, organic matter content 8 g kg^−1^, and available P, K, and total N concentrations were 8.6, 170, and 0.61 mg kg^−1^, respectively.

Ten seeds of fennel (*Foeniculum vulgare* Mill.) of Ardabil ecotype (this ecotype was grown in cool climates and was collected from the Ardabil area) were sown to the pots filled with the soil, and after seedlings were well established, they were thinned to five per pot. Using a pressure plate apparatus, volumetric soil moisture content at field capacity (FC) (−0.03 MPa) and permanent wilting point (PWP) (−1.5 MPa) were determined to be 0.42 and 0.092 g cm^−3^, respectively. Available soil water content is the difference between FC and PWP. Using a time-domain reflectometry or TDR (Moisture probe meter, ICT, MPM-160-B, USA) fitted with a 20 cm probe, soil moisture in each pot was checked twice every day, and each pot was watered as above by a sprinkler.

### 2.2. Experimental Design

Following standard protocol for experiments, treatments were arranged as a factorial in a randomized complete block design with three replicates. The levels of water stress were as follows: no stress (control), moderate, or severe stress. The used PGRs were: methyl jasmonate, 24-epibrassinolide, or putrescine (all reagents were obtained from Merck, Germany); the control plants were treated with distilled water.

Plants were grown at 80% FC (daily watering and checked by TDR) before the treatments were applied. In the control (non-stressed) plants, the FC was maintained at 80%, which was 0.32 ± 0.005 g cm^−3^ water per soil. Plants exposed to moderate or severe drought stress received 60% or 40%, respectively, of the amount of water applied to pots with the control plants. Water stress was started when one part of plants was in the vegetative state (65 days after planting) and when the other part was at the 50% generative stage (110 days after planting). The three PGR solutions (or water) were applied twice (by foliar spraying) at 3 and 5 days prior to the onset of stress treatments, and each time, 250 mL of PGRs solutions was applied to each pot. The PGRs were used at concentrations of: 24-epibrassinolide at 0.1 µM [31], putrescine at 0.5 mM [32], and methyl jasmonate at 50 μM [33,34]. The physiological and biochemical analyses of the plant material were carried out 20 days after establishing the differential water stress levels at vegetative or generative stages, whereas biomass and grain yield were determined when these two groups of drought-exposed and PGRs-treated plants have reached the generative stage (they were 160 days old).

### 2.3. Determination of Free Proline and Total Soluble Sugar Concentrations

Free proline level and total soluble sugars content were measured 20 days after starting of water deficit treatments, which has been shown to be the time of their highest accumulation after onset of stress [35]. Proline was extracted from 0.5 g of fresh leaves (full developed leaves (leafy part) from the top of the shoot) using the standard method developed by Bates et al. [36]. The content of this amino acid was determined spectrophotometrically (UV2100, Unico, Rosemount, Minnesota, USA) at 520 nm. Total soluble sugars were extracted from 0.5 g of fresh leaves (full developed leaves (leafy part) from the top of the shoot) using the method described by Irogoyen et al. [37], and sugar content was read at 625 nm using a spectrophotometer (UV2100, Unico, Rosemount, Minnesota, USA).

### 2.4. Analysis of Relative Water Content (RWC) in Leaves

For determination of RWC the leaf samples (full developed leaves from the top of the shoot) were allowed to float on distilled water for 4 h and then weighted to determine turgid mass (TM). Then, the plant material was oven-dried at 75 °C for 24 h and the dry weight (DW) was measured. RWC was calculated using the following formula:(1)$$\mathrm{RWC}=\frac{\mathrm{FM}-\mathrm{DM}}{\mathrm{TM}-\mathrm{DM}}\;\times \;100$$
where: FM, leaf fresh mass; DM, leaf dry mass; and TM, leaf turgid mass after floating on distilled water.

### 2.5. Biomass, Yield, and Yield Components

Grains, stems, and leaves were separated from the harvested plants and then dried at 75 °C for 24 h. The biomass of individual organs (shoots and grains) was determined using a laboratory balance. Five groups of 100 grains were weighted, and this value was used to calculate 1000-grain weight.

### 2.6. Extraction and Determination of Essential Oils

According to the methods of Zheljazkov et al. [38], 10 g of grains from each treatment were hydro-distilled for 3 h or until all the essential oils were removed, using a Clevenger apparatus. The extraction process continued until all essential oil was obtained. Total essential oil content was expressed as mg g^−1^ grain and essential oil yield as mg plant^−1^.

### 2.7. Calculation of Water Use Efficiency (WUE)

Plant water use efficiency (WUE) was calculated using the following formula [39]:(2)$$\mathrm{WUE}\;(\mathrm{unit}\;L^{-1})=\frac{\mathrm{plant}\;\mathrm{yield}\;}{\mathrm{water}\;\mathrm{consumed}}$$

### 2.8. Calculation of Spraying Use Efficiency (SUE) and Benefit to Cost Ratio (BCR)

Spraying use efficiency (SUE) of the PGRs was calculated using the following formula:(3)$$\mathrm{SUE}=\frac{{\mathrm{yield}}_{\mathrm{treatment}}-{\mathrm{yield}}_{\mathrm{control}}}{{\mathrm{cost}}_{\mathrm{treatment}}-{\mathrm{cost}}_{\mathrm{control}}}$$

The cost of spraying with PGRs was 1.06 × 10^−3^, 16.5 × 10^−3^, and 4.43 × 10^−3^ USD per pot for methyl jasmonate, epibrassinolide, and putrescine, respectively. Total irrigation water used for each pot at vegetative growth stage onwards was 25 L pot^−1^ in non-stressed treatment and 19 and 13 L pot^−1^ under moderate or severe stress treatments, respectively. Total amount of irrigation water used during the generative stage and onwards was 16.5, 12.9, and 10.6 L pot^−1^ in non-stressed, moderate, and severe stress, respectively. Total irrigation water applied before water treatment in the vegetative and 50% generative stages was about 10 and 18 L pot^−1^, respectively, and the price of each liter of water was determined based on 10×10^−5^ USD L^−1^. The benefit (product × price) to cost ratio (BCR) for grain and oil of fennel was calculated using the following formula [40]:(4)$$\mathrm{BCR}=\frac{{\mathrm{Benefit}}_{\mathrm{treatment}}-{\mathrm{Benefit}}_{\mathrm{control}}}{{\mathrm{cost}}_{\mathrm{treatment}}-{\mathrm{cost}}_{\mathrm{control}}}$$

### 2.9. Statistical Analysis

Data were subjected to analyses of variance (ANOVA) using SAS (SAS Inc., Cary, NC, USA). Least significant difference (LSD) was used to separate the means. A probability level of 0.05 was used to test the statistical significance among the treatments. Sigma plot v. 11 was used for drawing graphs.

## 3. Results

### 3.1. Free Proline and Total Soluble Sugar Levels

Proline content in fennel leaves was influenced by drought stress intensity at both developmental stages (Table 1), and as expected, water stress increased proline accumulation. Moderate or severe water stress during the vegetative stage increased proline concentration approx. two- and three-fold, respectively, compared with non-stressed plants (Table 1). However, when the flowering plants were exposed to water stress (generative stage) this increase was approx. 1.5-fold (Table 1). Total soluble sugars accumulation also increased under water stress; however, osmotic adjustment in fennel to maintain leaf turgidity during water stress was primarily through the accumulation of proline rather than total soluble sugars (Table 1). Plants during the vegetative and generative stages subjected to moderate water stress accumulated 1.6- and 1.5-fold more total soluble sugars, respectively, than the control (Table 1). Intensified water stress, however, did not result in further increases in total soluble sugar levels, and plants contained about 1.6 and 1.2 times more sugars than the control plants at the vegetative and generative stages, respectively (Table 1).

Overall, the application of PGRs did not stimulate the accumulation of free proline and soluble sugars under varying water regimes. Even under the influence of methyl jasmonate, a significant decrease in the level of proline and soluble sugars when applied during vegetative or generative phase, respectively, was found (Table 1). Moreover, there was no interaction effects between water stress and PGRs on the accumulation of compatible solutes, indicating that water stress increased proline and soluble sugars contents regardless of the application of PGRs (Table 1).

### 3.2. Relative Water Content in Leaves

Plants exposed to drought stress exhibited lower RWC than non-stressed plants (Table 1). However, the difference in RWC between control plants and those grown in stressful conditions was about 5% when averaged over the two stages of plant growth. The effect of PGRs application on RWC values was insignificant, and there was no interaction between PGRs and water stress (Table 1).

### 3.3. Biomass, Grain Yield, and Selected Yield Components

Water stress during the vegetative phase caused a substantial decrease in biomass, grain yield, and major grain yield components (Table 1 and Table 2). When plants were subjected to moderate or severe water stress, their biomass decreased by 37% and 60%, respectively, and the grain yield decreased by 27% and 61%, respectively (Table 2). Both grain yield components (grain number and weight of 1000-grain) were highly sensitive to limited irrigation. Severe drought decreased the number of grains per plant by 56%, and grains had 43% less weight than those produced by non-stressed plants. In contrast, flowering plants were not as strongly affected by drought as vegetative plants. However, grain yield and biomass of flowering plants were also decreased by 47% and 34%, respectively, in moderate water stress. Plants grown in severe water conditions had their grain and biomass yields, grain weight, and grain numbers reduced by about 60%, 42%, and 56%, respectively, compared with non-stressed plants.

The application of the three selected PGRs at both development stages stimulated both grain yield and biomass production. However, the increase in grain weight per plant was more pronounced than the increase in shoot biomass. Moreover, the magnitude of increase in grain and biomass yields depended on the time of the application of studied phytohormones. Application of PGRs during the vegetative stage had an impact on shoot biomass, and when averaged over the three PGRs, it was roughly 55% higher than that of the control plants (Table 2).

In contrast, the influence of the PGR applications during the generative stage on average resulted in only an 18% increase in the DW of vegetative organs, and the impact of the PGRs was not significant. On the other hand, the grain yield responded even more favorably than vegetative organs to the application of PGRs. Averaged over the three hormones, grain yield per plant was increased up to 2.5-fold when hormones were applied during the vegetative stage and 2.6-fold when were applied at generative stages. In general, methyl jasmonate and epibrassinolide were significantly more effective than putrescine in increasing grain production for both application times (Table 2). The application of methyl jasmonate, epibrassinolide, and putrescine during the vegetative stage increased the grain number per plant by 2.56-, 2.44-, and 1.45-fold, respectively. When plants in the generative stages were sprayed with the PGRs, their positive influence on the grain number per plant was a little less pronounced and exceeded the control value by 1.82-, 1.65-, and 1.52-fold, respectively. The application of all three PGRs also increased fennel grain weight (Table 2). Unlike the grain number, the grain weight benefited more from the PGRs being sprayed at the generative stages. Averaged over the three PGRs, the grain weights for plants at the vegetative and generative stages were 1.3- and 1.6-fold higher, respectively, than for the control plants.

The interaction between water stress level and the application of PGRs had a significant influence on grain yield only when PGRs were applied in the vegetative stage (Table 2 and Figure 1a–b). In these conditions, the highest grain production (2.7 g plant^−1^) was stated in the non-stressed plants treated with methyl jasmonate. In turn, under moderate water stress, the highest grain production (2.1 g plant^−1^) was found in plants treated with epibrassinolide. In the case of the two remaining PGRs, the increase in grain yield was clearly lower than in epibrassinolide-treated individuals, but still statistically significant (Figure 1a).

### 3.4. Essential Oil Content and Yield

Although water stress had a stimulatory influence on essential oil concentration in seeds (Table 2), the overall essential oil yield was decreased by 18% and 48% under moderate and severe stress starting in the vegetative phase, respectively, due to decreased grain number per plant. The essential oil yield was also decreased by 23% and 38% when plants were exposed to these stresses in the generative stages, respectively. In turn, the application of the PGRs before water stress exposition significantly improved the essential oil content of fennel grains at both stages of growth, being more pronounced in the vegetative stage (Table 3; Figure 2).

Grains contained 27% and 16% more total essential oil when plants were subjected to severe water deficit in the vegetative and reproductive stages, respectively, compared with those under well-watered conditions. The three PGRs used increased essential oil content, especially when plants were sprayed during the vegetative stage. Methyl jasmonate was the most effective among the applied PGRs in improving the essential oil content. Compared with the control plants (not treated with PGRs), the application of methyl jasmonate at the vegetative and generative stages increased the oil content of fennel grains by 26% and 16%, respectively (Table 3). Epibrassinolide had the highest beneficial impact on the essential oil yield when applied during the vegetative stage, whereas methyl jasmonate was the most favorable among the PGRs when plants were sprayed at the generative stage. The improvements in essential oil yield by these two PGRs were 4.2- and 3.3-fold for the vegetative and generative stages, respectively (Table 3).

There was a significant interaction between water stress and treatment with PGRs on grain yield, essential oil content, and yield for plants exposed to stress during the vegetative stage (Table 2 and Table 3). In these conditions, the maximum essential oil yield for well-watered plants treated with methyl jasmonate was 60 mg plant^−1^, while for plants grown under moderate water stress and sprayed with epibrassinolide, it was 50 mg plant^−1^ (Figure 1b).

### 3.5. Water Use Efficiency

Water deficit during either the vegetative or generative stage had a similar effect on the WUE of grain yield, while moderate water stress slightly improved the WUE index of grains. Plants that experienced severe stress during the vegetative or generative stage had 36% and 30% lower WUEs, respectively. A dramatic reduction in grain yield due to severe water deficit at both stages caused a significant decrease in the WUE. Compared with the WUE of grains, the reduction in the WUE of essential oil yield due to severe water stress was less, mainly due to the improved essential oil yield in stressful conditions (Table 3).

The PGRs, more specifically methyl jasmonate and epibrassinolide, improved both WUE values of grain and essential oil substantially when plants were sprayed at both stages of growth (Table 3). The application of PGRs before water stress exposition have a significant interaction effect on the WUE of grains and essential oils when the treatment occurred during the vegetative period (Figure 1). However, this interaction was not statistically significant during the generative stage. The highest WUE for grain (0.91 g L^−1^) and oil (20 mg L^−1^) was noted for well-watered plants treated with methyl jasmonate and for plants grown in moderate stress condition sprayed with epibrassinolide (0.80 g L^−1^ and 21 mg L^−1^ for grain and oil, respectively).

### 3.6. Analysis of Spraying Use Efficiency and Benefit-to-Cost Ratio

The spraying use efficiency (SUE) of grain and oil production was negatively affected by the exposure of fennel to drought stress. The highest SUE for grain and essential oil production was obtained for plants sprayed with methyl jasmonate; however, water stress dramatically decreased the positive effects of this PGRs on the SUE index. By spending USD 1 as the cost of methyl jasmonate application during the vegetative growth stage in non-stress conditions, there will be an increase of 1650 g plant^−1^ in grain and 42 g plant^−1^ in oil yield (Figure 3). Although epibrassinolide was the most effective of the three PGRs for most of the investigated traits, due to the high cost of application, it produced the lowest SUE in the non-stress and stressful conditions (Figure 3).

By spending USD 1 as the cost of methyl jasmonate application during the vegetative stage in non-stress condition, there was a USD 54 plant^−1^ increase in grain and USD 7.9 benefit from oil production. Furthermore, the application of PGRs during the generative stage of fennel led to an increase in the SUE in stress conditions. In addition, by spending USD 1 for the cost of methyl jasmonate, 13.2, 27.9, or 24.9 g increases in essential oil production under non-stress condition, moderate, or severe water deficit conditions, respectively, can be obtained. The results for the BCR revealed that for each cost of USD 1 for the application of methyl jasmonate during the generative stage under no-stress, moderate, or severe drought, there was a USD 27, USD 54, or USD 29 BCR in grain production, respectively, and a 2.7, 5.2, or 4.7 BCR in essential oil yield, respectively (Figure 3). Furthermore, as pointed out by a reviser, we need to keep in mind that the water use efficiency, spraying use efficiency (SUE), and benefit-to-cost ratio (BCR) are based on the experiment carried out in pots. These costs would be very different in field experiments, and determining the costs becomes even more complicated if the cost of the production of the chemicals (PGRs) is also considered.

## 4. Discussion

Fennel is generally considered a drought-sensitive species, although both drought-sensitive and drought-tolerant genotypes exist [27,41]. Based on our results, fennel also seems to have a low resistance to water deficit, and reduced availability of water during vegetative growth caused significant decreases in biomass and grain yields. Despite active osmotic adjustment and maintenance of the RWC, the biomass and grain yields of fennel showed a substantial decrease, regardless of the stage of plant development in which the stress occurred. Although water deficit led to a 40% saving in the cost of irrigation water, it decreased grain yield by approx. 60%.

Fennel plants have active osmotic adjustment and were capable of maintaining the RWC without the application of PGRs, confirming reports from earlier studies [41,42]. Due to osmotic adjustment, roots are able to take up water from the soil with low water content. The osmotic adjustment of fennel was primarily gained through a substantial increase in free proline concentration, which was higher when plants were subjected to water stress during the vegetative stage than at the generative stage. The results revealed that osmotic adjustment in fennel leading to maintenance of leaf turgidity and RWC during drought stress was primarily through accumulation of proline rather than total soluble sugars (Table 1). Proline, which accumulates during water and other abiotic stresses, plays a crucial role in maintaining proper water status in stressed plants and helps stabilize membranes and proteins structures, scavenge reactive oxygen species, alleviate cytoplasmic acidosis, and maintain appropriate NADP^+^/NADPH ratios, which are compatible with metabolism [43].

Unlike active osmotic adjustment, water stress caused serious decreases in yield and yield components, which varied with the water deficit intensity and plant growth stage. Fennel yield was affected by the weight and number of grains, and any reduction in these traits will led to yield loss. It has been reported that water stress can decrease grain formation in plants via four ways: (1) unsuccessful fertilization of the ovules during pollination; (2) reduction in the photosynthesis rate and assimilating supply from photosynthesis current [44], which can be supplemented with remobilized stem carbohydrate reserves during grain filling; (3) shortening of the grain filling duration under drought stress; and (4) the low/slow movement of assimilates to developing seeds [45] and leaves [46]. Water deficit during the vegetative stage can also decrease the time to anthesis, while water stress during the generative stage can cause a decrease in the grain-filling period, as, for example, in corn [47] and wheat [48].

The application of all three PGRs selected for our experiments alleviated the negative effects of water stress occurring during the vegetative period and thus significantly increased grain yield. However, when PGRs were applied to water-stressed plants in the generative stage, grain yield was not significantly increased. In non-stressed conditions, methyl jasmonate was the most efficient PGR in both stages, and its use resulted in the highest yield and most favorable yield components. In turn, in water deficit conditions, especially in the vegetative stage, epibrassinolide was the most effective PGR in alleviating the negative effect of stress (Figure 2). The amount of photosynthetic pigments and antioxidant activity increased in onion (*Allium cepa* L.) and *Arabidopsis thaliana* when methyl jasmonate was applied [49]. In turn, Wang et al. [46] found that the concentration of active cytokinins increased in rice treated with methyl jasmonate. Shahzadetal. [50] suggested that brassinosteroids can help alleviate oxidative stress by increasing rates of photosynthesis in treated plants. Thus, the positive effect of different PGRs on grain yield may be due to enhanced photosynthetic rates [51], improvements in assimilate partitioning [52], increased length of flowering period [53], and increased period of filling and/or grain growth rates [54].

The application of PGRs in the vegetative stage also increased the essential oil production and oil yield of fennel seeds, but this effect of the PGRs was less pronounced in the generative stage. Similar results have been reported in peppermint (*Mentha piperita* L.) [55]. If water for irrigation is limited, essential oil production by fennel seeds can be improved by moderate water stress but only if the stressed plants are treated with PGRs, especially epibrassinolide. However, the total yield of essential oils was seriously decreased due to the significant decrease in biomass of water-stressed plants, unlike reports on other aromatic plant species such as *Thymus carmanicus* [56].

It was hypothesized that fennel is a plant species with high WUE and that the application of PGRs can improve the WUE and grain production under water deficit, depending on the phase of plant development. However, any factor that causes an increase in yielding or a decrease in water used can increase the WUE [57]. The WUE of fennel was significantly improved by the application of the three PGRs used. This improvement in the WUE, especially at the vegetative stage, was mainly caused by less water consumption by plants, probably due to a decrease in the transpiration rate following the application of the PGRs. However, the detailed mechanisms of this beneficial effect are little known and, as our research shows, rather unrelated to the increased accumulation of free proline or soluble sugars.

The application of PGRs can increase plant tolerance to stress conditions, such as water deficit, and thereby improve plant growth and yielding [58,59]. Furthermore, the application of PGRs increased the production of secondary metabolites in some plant species [60,61], which has also been confirmed in our research. Moreover, it was found that the effects of methyl jasmonate and epibrassinolide on fennel plants were similar, but methyl jasmonate was more effective and economically beneficial to use than epibrassinolide. Putrescine was the least effective among the PGRs studied. Finally, it should be borne in mind that fennel is a species grown in the field, but preliminary laboratory or greenhouse pot experiments are a necessary step to test the benefits of application of various types of substances on plant growth under field conditions.

## 5. Conclusions

Fennel is a drought-sensitive medicinal species and exposition of plants to even moderate water deficit can cause a significant decrease in shoot biomass and grain yield. Our research showed that although water stress improved the concentration of essential oil in fennel grains, reduced irrigation alone is not a cost-effective strategy to use as an eliciting factor in fennel cultivation. However, when water availability is limited, moderate water stress could be employed to improve water use efficiency if plants are treated with PGRs during vegetative stage. Additionally, our study indicated the most effective PGRs in increasing plants’ resistance to water deficit and the phase of plant growth in which they should be applied. From an economic perspective, the application of methyl jasmonate can be the most favorable treatment under moderate water stress. However, the results of these studies should be verified under field conditions in the future, taking into account the total costs of treatments, including the availability and cost of chemicals, spraying equipment, or manpower.

## Figures and Tables

**Figure 1 plants-11-01718-f001:**
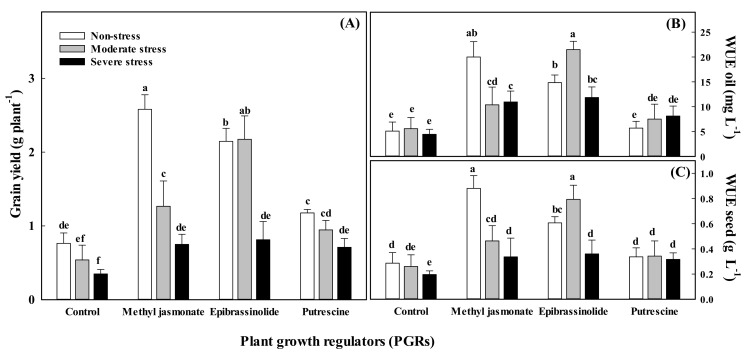
Interactive effect of plant growth regulators and water stress intensity on (**A**) grain yield, (**B**) water use efficiency for essential oil production, and (**C**) water use efficiency for seed yield of fennel (*Foeniculum vulgare* Mill.) at vegetative stage. Data are means (*n* = 3), and bars represent standard errors of the mean. In each panel, means with same letter (a, b, c, …) do not differ significantly (*p* < 0.05).

**Figure 2 plants-11-01718-f002:**
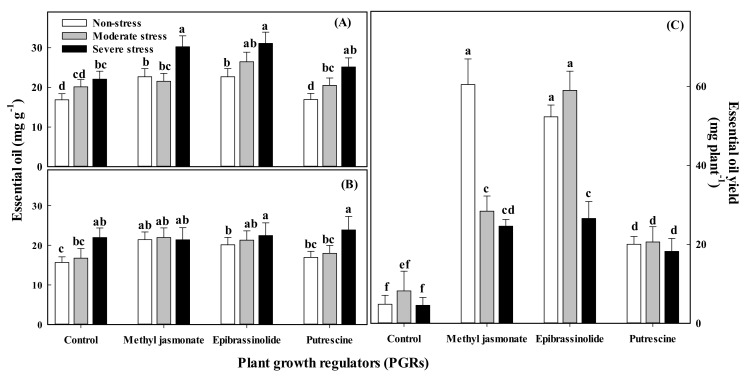
Effect of water stress and plant growth regulators applied at the (**A**) vegetative or (**B**) generative stage of fennel (Foeniculum vulgare Mill.) on essential oil content (**A**,**B**) and essential oil yield in vegetative stage (**C**). Data are means (*n* = 3), and bars represent standard errors of the mean. In each panel, means with same letter (a, b, c, …) do not differ significantly (*p* < 0.05).

**Figure 3 plants-11-01718-f003:**
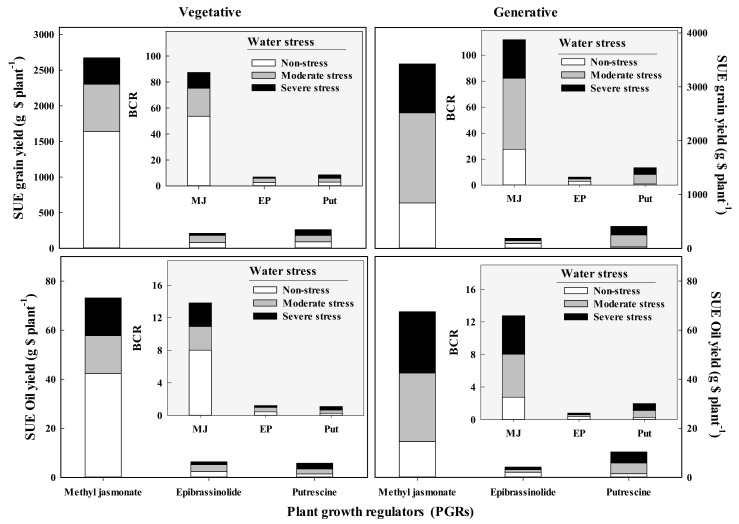
Spraying use efficiency (SUE) and benefit to cost ratio (BCR) of plant growth regulators on grain and essential oil yields of fennel (*Foeniculum vulgare* Mill.) grown under different water stress intensity during the vegetative or generative stage.

**Table 1 plants-11-01718-t001:** Influence of water stress and PGR applications on osmotic adjustment and RWC of fennel (*Foeniculum vulgare* Mill.) at two stages of development.

Treatments	Proline (µg g FW^−1^)		Total Soluble Sugars (mg g FW^−1^)		RWC (%)	
	Vegetative	Generative	Vegetative	Generative	Vegetative	Generative
Non-stress	^†^ 0.55 ± 0.06 b	1.32 ± 0.22 b	0.97 ± 0.08 b	0.36 ± 0.06 b	84.0 ± 1.6 a	81.5 ± 1.4 a
Moderate stress	1.39 ± 0.24 a	2.11 ± 0.21 a	1.55 ± 0.10 a	0.55 ± 0.05 a	80.1 ± 2.8 b	79.4 ± 2.4 b
Severe stress	1.58 ± 0.20 a	2.19 ± 0.25 a	1.60 ± 0.10 a	0.41 ± 0.05 ab	78.9 ± 3.0 b	77.9 ± 2.5 b
LSD (0.05)	0.50	0.53	0.19	0.14	2.69	2.69
PGRs						
Control	1.28 ± 0.20 ab	1.84 ± 0.31 b	1.64 ± 0.11 a	0.50 ± 0.03 a	79.9 ± 1.3 a	77.9 ± 1.2 a
Methyl jasmonate	0.74 ± 0.15 b	2.32 ± 0.29 a	1.46 ± 0.13 b	0.30 ± 0.06 c	80.8 ± 1.5 a	78.4 ± 1.5 a
Epibrassinolide	1.59 ± 0.35 a	2.03 ± 0.27 ab	0.95 ± 0.14 c	0.42 ± 0.10 b	82.8 ± 1.2 a	81.4 ± 0.9 a
Putrescine	1.09 ± 0.25 ab	1.30 ± 0.21 c	1.45 ± 0.12 b	0.54 ± 0.06 a	83.0 ± 1.1 a	80.7 ± 1.0 a
LSD (0.05)	0.58	0.61	0.22	0.16	3.1	3.1
^††^ F test						
Water stress (WS)	**	**	**	*	**	*
PGR	*	*	**	*	ns	ns
WS × PGR	ns	ns	ns	ns	ns	ns
CV (%)	15.6	17.9	16.8	23.5	3.89	4.65

^†^ Data are means and ± standard error of the mean. In each panel, means with same letter (a, b, c,…) do not differ significantly (*p* < 0.05). ^††^ ns, *, ** are: not significant, *p* ≤ 0.05, and *p* ≤ 0.01, respectively.

**Table 2 plants-11-01718-t002:** Influence of water stress and PGRs application on grain yield and selected yield components of fennel (*Foeniculum vulgare* Mill.) at two stages of development.

Treatments	Grain Number (per Plant)		1000-Grain wt. (g)		Grain Yield (g Plant^−1^)		Shoot Biomass (g Plant^−1^)	
	Vegetative	Generative	Vegetative	Generative	Vegetative	Generative	Vegetative	Generative
Non-stress	^†^ 539.6 ± 99.0 a	529.2 ± 104.1 a	3.17 ± 0.26 a	3.25 ± 0.40 a	1.68 ± 0.32 a	1.72 ± 0.31 a	13.5 ± 0.95 a	13.8 ± 0.60 a
Moderate stress	459.2 ± 117.0 b	470.0 ± 98.2 ab	2.28 ± 0.38 ab	3.11 ± 0.32 a	1.23 ± 0.22 a	1.43 ± 0.26 a	8.6 ± 0.81 b	13.8 ± 0.90 a
Severe stress	237.3 ± 48.1 c	389.1 ± 77.4 b	1.82 ± 0.30 b	2.01 ± 0.21 b	0.66 ± 0.11 b	0.91 ± 0.20 b	5.6 ± 0.27 c	9.2 ± 1.70 b
LSD (0.05)	241.0	228.7	0.85	0.73	0.37	0.68	1.7	3.6
PGRs								
Control	220.6 ± 35.4 c	307.5 ± 65.5 c	2.01 ± 0.18 c	1.92 ± 0.14 c	0.55 ± 0.11 c	0.61 ± 0.10 c	6.3 ± 0.9 b	10.8 ± 1.16 a
Methyl jasmonate	566.1 ± 104.1 a	561.8 ± 106.1 a	2.20 ± 0.43 ab	3.44 ± 0.50 a	1.53 ± 0.34 a	1.93 ± 0.29 ab	9.3 ± 1.5 a	14.9 ± 1.39 a
Epibrassinolide	540.2 ± 105.1 ab	509.4 ± 139.2 ab	3.35 ± 0.39 a	3.29 ± 0.38 a	1.71 ± 0.36 a	1.65 ± 0.33 a	9.9 ± 1.4 a	11.8 ± 1.24 a
Putrescine	320.3 ± 81.2 b	469.8 ± 99.8 b	2.41 ± 0.44 b	2.79 ± 0.24 b	0.94 ± 0.14 b	1.23 ± 0.32 b	10.1 ± 1.7 a	11.8 ± 1.12 a
LSD (0.05)	248.3	239.5	0.98	0.85	0.42	0.79	1.95	4.2
^††^ F test								
Water stress (WS)	*	ns	*	*	**	*	**	**
PGR	*	*	*	**	**	*	**	ns
WS × PGR	ns	ns	ns	ns	*	ns	ns	ns
CV (%)	16.90	19.1	18.2	13.2	19.62	20.15	13.15	15.6

^†^ Data are means and ± standard error of the mean. In each panel, means with same letter (a, b, c, …) do not differ significantly (*p* < 0.05). ^††^ ns, *, ** are: not significant, *p* ≤ 0.05, and *p* ≤ 0.01, respectively.

**Table 3 plants-11-01718-t003:** Influence of water stress and PGRs application on essential oil content and water use efficiency of fennel (*Foeniculum vulgare* Mill.) at two stages of the development.

Treatments	Essential Oil Content (mg g^−1^ of Seeds)		Essential Oil Yield (mg Plant^−1^)		WUE for Grain (g L^−1^)		WUE for Essential Oil (mg L^−1^)	
	Vegetative	Generative	Vegetative	Generative	Vegetative	Generative	Vegetative	Generative
Non-stress	^†^ 19.79 ± 1.2 b	18.53 ± 1.0 b	36.6 ± 8.7 a	34.6 ± 6.0 a	0.47 ± 0.09 ab	0.46 ± 0.06 ab	11.4 ± 2.5 a	11.04 ± 1.2 a
Moderate stress	22.16 ± 1.1 b	20.79 ± 1.1 a	29.9 ± 6.7 b	26.7 ± 5.1 ab	0.53 ± 0.08 a	0.54 ± 0.05 a	11.3 ± 2.4 a	12.23 ± 1.5 a
Severe stress	27.13 ± 1.5 a	21.11 ± 1.5 a	19.3 ± 4.0 c	21.4 ± 5.3 b	0.30 ± 0.05 b	0.32 ± 0.06 b	8.9 ± 1.7 a	9.56 ± 0.6 a
LSD (0.05)	0.60	1.43	9.8	12.5	0.12	0.13	3.02	3.12
PGRs								
Control	19.69 ± 1.1 c	18.12 ± 1.4 b	11.0 ± 2.3 c	11.5 ± 2.3 b	0.25 ± 0.04 b	0.24 ± 0.04 c	5.1 ± 0.9 b	5.35 ± 0.5 c
Methyl jasmonate	24.82 ± 1.8 ab	19.59 ± 1.6 b	37.8 ± 8.6 b	37.3 ± 5.9 a	0.56 ± 0.10 a	0.55 ± 0.06 a	13.8 ± 2.5 a	14.94 ± 1.5 a
Epibrassinolide	26.74 ± 1.7 a	21.58 ± 1.2 a	45.9 ± 9.5 a	34.6 ± 5.8 a	0.59 ± 0.11 a	0.53 ± 0.06 a	16.1 ± 2.9 a	12.88 ± 1.3 ab
Putrescine	20.86 ± 1.5 bc	21.28 ± 1.3 a	19.6 ± 3.0 c	26.9 ± 7.4 a	0.33 ± 0.04 b	0.44 ± 0.07 b	7.1 ± 1.2 b	10.60 ± 1.7 bc
LSD (0.05)	0.77	1.65	11.3	15.7	0.13	0.15	3.5	3.6
^††^ F test								
Water stress (WS)	**	**	**	*	**	*	ns	ns
PGR	**	**	**	**	**	*	**	*
WS × PGR	**	**	*	ns	*	ns	*	ns
CV (%)	3.11	8.38	18.2	13.2	18.65	20.2	18.5	19.2

^†^ Data are means and ± standard error of the mean. In each panel, means with same letter (a, b, c, …) do not differ significantly (*p* < 0.05). ^††^ ns, *, ** are: not significant, *p* ≤ 0.05, and *p* ≤ 0.01, respectively.

## Data Availability

Not applicable.

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
