# Peer review of "Plant Growth Regulators Improve Grain Production and Water Use Efficiency of Foeniculum vulgare Mill. under Water Stress"

_plants, 2022, doi:10.3390/plants11131718_

Round 1

Reviewer 1 Report

This article talk about improving of water stress by PGRs application in funnel which one of useful spice. It is rare of research studying on spice. This work is valuable and can be publish after revision in many points as following;

-Title should be changes to shoter and concise.

-Introduction: Authors should added more details .....
1) Why methy-jasmonate, epibrassinolide and putrescine were selected to apply in this study?
2) Lines 50-72 were separated into 2 paragraphes. Authors should revised by mixing them to a paragraph or spared in to 3 paragraphes by types of PGR.
3) Lines 73-81: Authors wrote down that aromatic plants are tolerate to water stress. It has to describe that why this work have to do about funnel which a aromatic plant under water stress. Since, funnel might tolerate to water stress from literature review.

-Materials and mathods
1) Line 114: What is FC and how to control to 80%FC?
2) Line 115: Is it the number "0.32 g/cm" of water or planting material?

-Results
1) Every tables should as SD or SE. Only CV is not enough to reflect your data.
2) Every tables should as statistic description as a, b, c because they already analyzed by pos-hoc LSD at 0.05.

Lastly, grammarly editing by native speaker is recommended before publish.

Author Response

Author's Reply to the Review Report (Reviewer 1)

Dear Reviewer,

Thank you very much for your comments, which have been considered and incorporated in the new version of the manuscript. The detailed list of responses is given below. We hope that the modifications and explanation will be will be sufficient to accept the manuscript for publication.

Comments and Suggestions for Authors

This article talk about improving of water stress by PGRs application in funnel which one of useful spice. It is rare of research studying on spice. This work is valuable and can be publish after revision in many points as following;

- Title should be changes to shoter and concise.

Answer: The title was improved to ‘Plant growth regulators improve grain production and water use efficiency of Foeniculum vulgare Mill. under water stress’

-Introduction: Authors should added more details .....
1) Why methy-jasmonate, epibrassinolide and putrescine were selected to apply in this study?

Answer: This information has been placed at the end of the second paragraph of the introduction.

2) Lines 50-72 were separated into 2 paragraphes. Authors should revised by mixing them to a paragraph or spared in to 3 paragraphes by types of PGR.

Answer: These corrections have been made.

3) Lines 73-81: Authors wrote down that aromatic plants are tolerate to water stress. It has to describe that why this work have to do about funnel which a aromatic plant under water stress. Since, funnel might tolerate to water stress from literature review.
Answer: These corrections have been taken into account and the relevant information has been provided in the pre-last paragraph of the introduction.

-Materials and mathods
1) Line 114: What is FC and how to control to 80%FC?

2) Line 115: Is it the number "0.32 g/cm" of water or planting material?

Answer: These corrections have been made.

-Results
1) every tables should as SD or SE. Only CV is not enough to reflect your data.

2) Every tables should as statistic description as a, b, c because they already analyzed by pos-hoc LSD at 0.05.

Answer: These corrections have been made.

Reviewer 2 Report

In this paper the effects of water deficit in different growth stages of fennel on biomass and seed yield, essential oil content and water use efficiency is examined as well as the ameliorating effects of three PGRs methyl jasmonate, epibrassinolide, and putrescine.

Generally, the paper is well written and comprehensively descriptive with respect to most sections.

General comments:

The terms “Generative stages / Reproductive / Flowering phase / Flowering stage” are used. Consider using one for consistency.

This is a detailed and comprehensive study but the interpretation and the significance of the pot experiments with respect to fennel as a field crop should be considered and commented on in the relevant sections.

Outline more specifically the significance of choosing fennel with respect to agricultural practices in Iran.

Water use efficiency, spraying use efficiency (SUE) and benefit to cost ratio (BCR) are based on the experiment carried out in pots. These costs would be very different in field experiments. This should be discussed in the paper. It becomes more complicated if the cost of the production of the chemicals (PGRs) is also considered.

Some points are covered more specifically below:

Title

“Plant growth regulators improve grain production and water use efficiency of Foeniculum vulgare Mill. in moderate but not in severe water stress conditions”  

Lines 7, 9, 17.  1 Department of Agronomy, Faculty of Agriculture, University of Mohaghegh Ardabili, Ardabil, Iran;

This address is the same for three authors so 1 could be applied also to Ali Ebadi2 and Soodabe Jahan-bakhsh6. Accordingly, the other address numbers will change also. 

Abstract

The abstract comprehensively covers the study.

L. 21 maybe use “Aromatic and medicinally…’

L. 23 “The selected PGRs, e.g. methyl jasmonate, epibrassinolide, or putrescine were applied to water-stressed fennel at the vegetative and 50% flowering stages”. E.g. not necessary, applied by spraying 

Keywords: Consider adding: water stress, Foeniculum vulgare L.

Introduction

The introduction is generally comprehensive of the topic and lays out the objectives of the study. However, it could be supplemented with information on why the authors chose to work with fennel. It could be considered to be used more often in the food industry than for its medical properties. It would ordinarily be a non-irrigated rain-fed crop in field cultivation. Is it a new/major crop for Iran?

Line 55. Replace with this ‘These phytohormones enhance..’

Materials and Methods

The Materials and Methods are generally well described. Some comments below:

L. 95 “maximal photosynthetic photon flux density (PPFD) was about 700 μmol m-2 s-1.” Was this ambient sunlight or were lamps used? And which season/month was the experiment conducted?

L. 101 “Ardabil ecotype” Give more information about this genotype eg origin, cultivation

L. 105 “Available soil water content is the difference between FC and PWP. Using a TDR (Moisture probe meter, ICT, MPM-160-B, USA) fitted with a 20 cm probe, soil moisture in each pot was checked twice every day”. How were pots watered? From above / drip?

L. 134 “0.5 g of fresh leaves (full developed leaves from the top of the shoot)”. Consider “most recent fully-developed leaves”. Did this sample include the petiole stem or just the leafy part?

Results

Table 1. left-align treatment column

Table 2. species name in italics

Why is there such a large difference in the values of the parameters between the non-stress treatment and the control? Explain this.

Figure 1. species name in italics

Figure 2. species name in italics. Use A, B, C for figure parts so as not to confuse with statistical a, b etc

L 302. “(a) vegetative or (b) reproductive stages of fennel” Elsewhere “generative” used

Table 3. species name in italics. Left-align treatment column.

Why is there such a large difference in the values of the parameters between the non-stress (36.6) treatment and the control (11.0) for Essential oil yield (mg plant-1) and WUE for essential oil (mg L-1). Explain this.

Columns “WUE for grain (g L-1) and WUE for essential oil (mg L-1)” Is this the same data given in Figure 1.?

Figure 3. species name in italics. Check the spaces between words on the axes eg SUE Grain yield. Label axes in insert figures at least with abbreviations for PGRs.

L 219. “However, grain yield and biomass of flowering plants were also decreased by 47% and 34%, respectively”. Correct if this refers to moderate water stress?

Discussion

The discussion needs to include the significance of the pot experiments with respect to fennel as a field crop (see general comments). This should be considered and commented on.

Conclusions

Authors should consider the feasibility of using such treatments with PGRs in field grown fennel. Field grown fennel is usually only rain-fed and not irrigated. In terms of costs, availability of chemicals, spraying equipment etc

References

Check that species names are in italics Lines 448, 458, 461, 466, 471, 480, 489. 495, 499, 501, 508, 516, 521, 532, 534, 537, 541, 543, 546, 558, 560, 563

Check that species names have lower case eg L 458 graveolens, L 480 daenensis see also L 489, 495, 499, 501, 508, 516, 521, 532m 534, 537, 541, 543, 546, 558, 560, 563

L 484 Menthapulegium space / italics Mentha pulegium

Reference 26 no year given

Reference 33 no volume/ page given

Author Response

Author's Reply to the Review Report (Reviewer 2)

Dear Reviewer,

Thank you very much for your comments, which have been considered and incorporated in the new version of the manuscript. The detailed list of responses is given below. We hope that the modifications and explanation will be will be sufficient to accept the manuscript for publication.

Comments and Suggestions for Authors

In this paper the effects of water deficit in different growth stages of fennel on biomass and seed yield, essential oil content and water use efficiency is examined as well as the ameliorating effects of three PGRs methyl jasmonate, epibrassinolide, and putrescine. Generally, the paper is well written and comprehensively descriptive with respect to most sections.

General comments:

The terms “Generative stages / Reproductive / Flowering phase / Flowering stage” are used. Consider using one for consistency.

Answer: These corrections have been made.

This is a detailed and comprehensive study but the interpretation and the significance of the pot experiments with respect to fennel as a field crop should be considered and commented on in the relevant sections.

Answer: These corrections have been taken into account and the relevant information has been provided in the last paragraph of the discussion and conclusion section.

Outline more specifically the significance of choosing fennel with respect to agricultural practices in Iran.

Answer: The appropriate information has been incorporated in the pre-last paragraph of the introduction.

Water use efficiency, spraying use efficiency (SUE) and benefit to cost ratio (BCR) are based on the experiment carried out in pots. These costs would be very different in field experiments. This should be discussed in the paper. It becomes more complicated if the cost of the production of the chemicals (PGRs) is also considered.

Answer:  This is an excellent point, and we have added your thoughts about differences between costs of pot and field studies to the discussion.

Some points are covered more specifically below:

Title

“Plant growth regulators improve grain production and water use efficiency of Foeniculum vulgare Mill. in moderate but not in severe water stress conditions”  

Lines 7, 9, 17.  1 Department of Agronomy, Faculty of Agriculture, University of Mohaghegh Ardabili, Ardabil, Iran;

This address is the same for three authors so 1 could be applied also to Ali Ebadi2 and Soodabe Jahan-bakhsh6. Accordingly, the other address numbers will change also. 

Answer: These corrections have been made.

Abstract

The abstract comprehensively covers the study.

  1. 21 maybe use “Aromatic and medicinally…’
  2. 23 “The selected PGRs, e.g. methyl jasmonate, epibrassinolide, or putrescine were applied to water-stressed fennel at the vegetative and 50% flowering stages”. E.g. not necessary, applied by spraying

Answer: These corrections have been made.

Keywords: Consider adding: water stress, Foeniculum vulgare L.

Answer: This has been done.

Introduction

The introduction is generally comprehensive of the topic and lays out the objectives of the study. However, it could be supplemented with information on why the authors chose to work with fennel. It could be considered to be used more often in the food industry than for its medical properties. It would ordinarily be a non-irrigated rain-fed crop in field cultivation. Is it a new/major crop for Iran?

Line 55. Replace with this ‘These phytohormones enhance..’

Answer: This change has been made.  

Materials and Methods

The Materials and Methods are generally well described. Some comments below:

  1. 95 “maximal photosynthetic photon flux density (PPFD) was about 700 μmol m-2 s-1.” Was this ambient sunlight or were lamps used? And which season/month was the experiment conducted?

Answer: These corrections have been made.

  1. 101 “Ardabil ecotype” Give more information about this genotype eg origin, cultivation

Answer: This has been done.

  1. 105 “Available soil water content is the difference between FC and PWP. Using a TDR (Moisture probe meter, ICT, MPM-160-B, USA) fitted with a 20 cm probe, soil moisture in each pot was checked twice every day”. How were pots watered? From above / drip?

Answer:. This information has been added.

  1. 134 “0.5 g of fresh leaves (full developed leaves from the top of the shoot)”. Consider “most recent fully-developed leaves”. Did this sample include the petiole stem or just the leafy part?

Answer: These corrections have been made.

Results

Table 1. Left-align treatment column

Table 2. Species name in italics

Why is there such a large difference in the values of the parameters between the non-stress treatment and the control? Explain this.

Figure 1. Species name in italics

Figure 2. Species name in italics. Use A, B, C for figure parts so as not to confuse with statistical a, b etc

L 302. “(a) Vegetative or (b) reproductive stages of fennel” Elsewhere “generative” used

Table 3. Species name in italics. Left-align treatment column.

Answer: These corrections/changes have been made.

Figure 3. Species name in italics. Check the spaces between words on the axes eg SUE Grain yield. Label axes in insert figures at least with abbreviations for PGRs.

Answer: These corrections have been made.

L 219. “However, grain yield and biomass of flowering plants were also decreased by 47% and 34%, respectively”. Correct if this refers to moderate water stress?

Answer:. These corrections have been made.

Discussion

The discussion needs to include the significance of the pot experiments with respect to fennel as a field crop (see general comments). This should be considered and commented on.

Answer: These suggestions have been taken into account and the relevant information has been provided in the last paragraph of the discussion.

Conclusions

Authors should consider the feasibility of using such treatments with PGRs in field grown fennel. Field grown fennel is usually only rain-fed and not irrigated. In terms of costs, availability of chemicals, spraying equipment etc

Answer: These suggestions have been taken into account and the relevant information has been provided in the last paragraph of the conclusion section.

References

Check that species names are in italics Lines 448, 458, 461, 466, 471, 480, 489. 495, 499, 501, 508, 516, 521, 532, 534, 537, 541, 543, 546, 558, 560, 563

Check that species names have lower case eg L 458 graveolens, L 480 daenensis see also L 489, 495, 499, 501, 508, 516, 521, 532m 534, 537, 541, 543, 546, 558, 560, 563

L 484 Menthapulegium space / italics Mentha pulegium

Reference 26 no year given

Reference 33 no volume/ page given

Answer: These corrections have been made.

Round 2

Reviewer 1 Report

It has a little minor comments as following;

1. Please give full name of abbreviation-PWP and TDR.

2. P.11, line 436: change "of course" to "finally", line 438 omit "(or not)

Author Response

Dear Reviewer,

Thank you very much for your comments, which have been considered and incorporated in the new version of the manuscript. The detailed list of responses is given below. We hope that the modifications and explanation will be will be sufficient to accept the manuscript for publication.

Academic Editor Notes

Lines 8, 10 and 12: cancel the hyphens right after the superscript numbers

Answer: These corrections have been made.

L8-9: use a comma after "University of Massachusetts"

Answer: These corrections have been made.

L56: use "the" before "regulation" and "of" after that ("the regulation of")

Answer: These corrections have been made.

L335: use "of" after "Analysis" in the title of the subsection

Answer: These corrections have been made.

L458: cancel the term "Please add:"

Answer: These corrections have been made.

References (for example L474, 484 and others): binomial names should be written in italics (this is OK) and capitalised only in the first letter of only the genus name. So, species names are not capitalised - please correct all.

Answer: These corrections have been made.

Author's Reply to the Review Report (Reviewer 1)

Comments and Suggestions for Authors

It has a little minor comments as following;

  1. Please give full name of abbreviation-PWP and TDR.
  2. P.11, line 436: change "of course" to "finally", line 438 omit "(or not)

Answer: These corrections have been made.
